# Connecting the Evolution and Spread of Turkey Reovirus Across the United States: A Genomic Perspective

**DOI:** 10.3390/v17091185

**Published:** 2025-08-29

**Authors:** Nakarin Pamornchainavakul, Jonathan T. Vannatta, Vikash K. Singh, Robert Porter, Sagar M. Goyal, Sunil K. Mor, Kimberly VanderWaal

**Affiliations:** 1Department of Veterinary Population Medicine, College of Veterinary Medicine, University of Minnesota, Twin Cities, MN 30471, USA; pamor001@umn.edu (N.P.); vsingh@umn.edu (V.K.S.); porte349@umn.edu (R.P.); goyal001@umn.edu (S.M.G.); 2College of Science, Engineering and Technology, Minnesota State University, Mankato, MN 56001, USA; jonathan.vannatta@mnsu.edu; 3Animal Disease Research and Diagnostic Laboratory, Department of Veterinary and Biomedical Sciences, College of Agriculture, Food & Environmental Sciences, South Dakota State University, Brookings, SD 57007, USA

**Keywords:** turkey reovirus, phylogeography, evolution, whole-genome analysis, epidemic, segmented virus

## Abstract

A major cause of lameness in turkeys is reoviral arthritis, driven by turkey reovirus (TRV) infection. In the U.S., TRV was first isolated in the 1980s but re-emerged as a significant pathogen causing arthritis in 2011. Since then, TRV outbreaks have spread nationwide across turkey-producing regions and have occasionally resulted in hepatitis-associated pathotypes. The absence of a consistently effective commercial vaccine continues to hinder disease control efforts. To better understand TRV’s evolutionary trajectory and transmission dynamics, we analyzed 211 complete TRV genome sequences collected across the U.S. from 2007 to 2021. Bayesian time-scaled phylogenetic and phylogeographic analyses were conducted for all ten genome segments to estimate gene flow among geographic regions, client groups, and pathotypes. The results reconstructed a coherent, decades-long history of TRV evolution, which revealed segment-specific differences in the evolutionary rates—particularly in *S1c* (σC protein coding region of *S1*) and *M2*—suggesting reassortment with other avian reoviruses during the 2011 emergence. Nationwide spread patterns indicated dominant transmission routes from the Eastern U.S. to Minnesota and from breeder to smallholder flocks, likely driven by inter-regional animal or feed movement via the multi-stage turkey production cycle. Pathotype transitions were more frequently observed from arthritis-associated strains to those causing hepatitis or cardiac lesions. These findings provide crucial insights to support national TRV control strategies and long-term monitoring by industry stakeholders.

## 1. Introduction

Since 2011, turkey reovirus (TRV) has been recognized as an emergent virus in the U.S., associated with lameness, a major health concern leading to high morbidity and mortality in turkeys [1,2]. Highly pathogenic turkey arthritis reovirus (TARV, or TRV when isolated from arthritis cases) has been estimated to cost the U.S. turkey industry up to USD 33.7 million as of 2019 [3]. Historically, reoviruses associated with turkey arthritis were isolated in the 1980s but then appeared to be absent from the field until the emergence of TARV in 2011 [3]. Apart from the emerging TARV pathotype, TRV is sometimes associated with other forms of pathogenesis, including turkey enteric reovirus (TERV) and turkey hepatitis reovirus (THRV), although the association between such pathotypes and viral genotype is still unclear [4].

TRV is an avian reovirus (ARV) belonging to the genus *Orthoreovirus*. Generally, ARVs consist of ten double-stranded RNA gene segments, which are classified into three size-based classes: three long (*L1*, *L2*, *L3*), three medium (*M1*, *M2*, *M3*), and four short (*S1*, *S2*, *S3*, *S4*) segments. These segments primarily encode three λ proteins (λA, λB, λC), three µ proteins (µA, µB, µNS), and four σ proteins (σC, σA, σB, σNS), respectively [5,6,7]. The *S1* gene also encodes two other non-structural proteins, namely, p10 and p17 [8]. All gene segments have been utilized to characterize, classify, or compare TRVs among themselves or with other ARVs [1,9,10,11,12,13,14,15], with the extent of usage depending on their functional significance or the variability of each gene. For example, σC, one of the products of the *S1* gene widely employed for ARV classification [16], is the most variable protein [17,18], which mediates virus-cell attachment [19], induces cell apoptosis [20], and is a target of neutralizing antibodies [21,22]. Six ARV genotypes have been described based on the *S1* gene or σC amino acid phylogenies [17], and most reoviruses found in turkeys have been classified as genotype 2 (all samples in this study are ARV genotype 2 according to σC phylogeny (Appendix A)) [9,13,23]. However, phylogenetic clusters of TRVs, as well as other ARVs, are incongruent among trees created from the ten segments, suggesting potential reassortment, i.e., the shuffling of gene segments between viruses infecting the same cell [12,13,14,15,23].

For over a decade since its re-emergence, TRV has continuously circulated, posing a threat to the nationwide turkey industry. An outbreak was first reported in the early 2010s in both Midwestern and Eastern states [1,9,24]. Since then, the disease has spread to every turkey-raising area in the U.S. [25] According to a National Turkey Federation Survey, there was more than a 100% increase in TRV cases from 2018 to 2019 [3], and the increasing trend of TARV-positive cases submitted to the University of Minnesota Veterinary Diagnostic Laboratory (MVDL) [26] underscores the unsuccessful control and prevention of the disease thus far. In the face of limited protection offered by vaccines, partly due to the continuous emergence of viral variants [27], there is a need for foundational knowledge on the rate at which the TRV evolves and reassorts. In addition, genomic epidemiology can provide insights into the national-scale epidemiology of TRV, especially the spatial and temporal dynamics of disease spread, which is an essential first step to developing control strategies to eliminate this virus.

Here, we utilized TRV whole-genome sequencing data from cases collected over 15 years from major turkey-raising regions in the U.S. and Canada, which were submitted to MVDL. We employed these data to estimate the virus’s evolutionary history, population dynamics, and patterns of geographical spread through phylodynamic analyses. We also addressed and summarized the complexity of historical inference due to reassortment by analyzing all TRV gene segments. Insights from our study represent a crucial first step in shedding light on the vague origin and dynamics of TRV and will aid in the management of future outbreaks.

## 2. Methods

### 2.1. Data Preparation

Data for this study came from the Minnesota Veterinary Diagnostic Laboratory (MVDL). Regardless of pathotype, all available complete genome sequences from avian reovirus (ARV) isolated from turkey cases across the U.S. and Canada were initially included in this study (*n* = 257). RNA extraction and sequencing varied over time but generally involved non-targeted next generation sequencing (NGS), using Illumina Miseq and NextSeq sequencers (Illumina, San Diego, CA, USA). The sources of sequences were grouped into seven geographic regions: Canada (*n* = 59); Dakotas (ND, SD; *n* = 21); Midwest (KS, MO, AR; *n* = 20); Upper Midwest (IA, WI; *n* = 17); Minnesota (MN; *n* = 87); North Central (MI, IN, OH, KY; *n* = 12); and East (PA, VA, NC, SC; *n* = 35). This grouping was based on turkey density, state adjacency, and sequence availability. Some sequences lacked geographic information and were therefore excluded from regional grouping. The Midwestern states were divided into three regions based on sample availability.

Sequences of each segment were aligned using MAFFT v7.5.20 [28] with the automatically selected alignment strategy. The 5′ untranslated regions (UTR) and 3′ UTR were trimmed from each alignment, except for the *S1* gene, in which only the open reading frame 3 (*ORF3*) portion encoding the σC protein (referred to as *S1c* in this paper) was retained due to high missing data in *ORFs 1* and *2*. Preliminary maximum likelihood phylogenetic trees were built from each alignment using IQ-TREE v2.3.0 [29], with the best-fit nucleotide substitution model selected by ModelFinder [30] and 1000 ultrafast bootstrap [31] trees. The resulting consensus trees, along with sampling dates, were then used to assess the temporal signal via TempEst v1.5.3 [32]. Potential recombination was also investigated in every alignment using RDP v4.101 [33].

Sequences that met at least one of the following criteria were removed from the alignment: (1) sequences without date or state-level location data, including a single sequence from California that was excluded because it alone was not sufficient to represent that region (*n* = 6); (2) genome sequences that were 100% identical and were collected from the same client in the same geographic region, and within the same calendar year (*n* = 20); (3) outlier sequences visually identified in at least one gene based on scatter plots between the root-to-tip divergence versus time generated by TempEst (*n* = 4); and (4) recombinant sequences in each alignment detected by more than four out of seven detection methods in RDP (*n* = 16). The final dataset for further analyses consisted of 211 samples from 16 states and Canada collected between 2007–2021 (Appendix A). These samples were provided by 30 production companies or clients, which were regrouped and deidentified into six major clients: a, b, c, d, e, and u (with u representing all clients that each contributed less than 5% of the total samples). The best-fit substitution models and temporal signals (Appendix A) were reevaluated by repeating the analyses conducted by IQ-TREE and TempEst on all gene alignments of this final dataset.

### 2.2. Phylodynamic Analysis

Bayesian phylodynamic analyses were performed using the final sequence alignment of all ten TRV gene segments as input to estimate the evolutionary dynamics and patterns of gene flow across different traits. For each alignment, the sampling date and the best-fit substitution model, determined by IQ-TREE’s ModelFinder [30] (Appendix A), were primarily utilized along with the uncorrelated relaxed clock model [34] with a log-normal distribution to reconstruct a time-scaled phylogenetic tree. Meanwhile, we estimated the frequency of virus transitions between different sample traits, including geographical regions (*n* = 7), clients (*n* = 6), and pathotypes (*n* = 4; TARV, THRV, TERV, and TRV for samples isolated from the heart or spleen, which are not typical sites of lesions). We used a non-reversible continuous-time Markov chain (CTMC) for the asymmetric discrete trait transition rate model with Bayesian Stochastic Search Variable Selection (BSSVS) to retain only the iterations that sufficiently explain the phylogenetic diffusion process [35,36]. The inference of these transitions was expected to reflect spread patterns and hotspots between geographical regions, major clients, and patterns of pathotype shift, respectively. The temporal population dynamics of TRVs were inferred from their gene segments using the Bayesian Skygrid’s Gaussian Markov random field (GMRF) model [37]. All Bayesian analyses were performed simultaneously with a Markov chain Monte Carlo (MCMC) chain length of 300 million, via BEAST v.1.10.4 [38].

After the analysis, the results were inspected using Tracer v1.7.2 [39]. For each gene segment, an appropriate burn-in was determined based on achieving effective sample sizes (ESS) > 100 for key parameters (see Appendix A), and the corresponding MCMC states were discarded before further analysis. Time-scaled trees and other parameters of the remaining states were summarized using TreeAnnotator v.1.10.4 [40] and SpreaD3 v.0.9.6 [41] into a maximum clade credibility (MCC) tree. The median time to the most recent common ancestor (tMRCA), median substitution rate, median effective population size over time, and the absolute rate of between-trait transitions with corresponding Bayes factors (BFs) were extracted from model outputs. We visualized time-scaled phylogenetic trees for the *M2* and *S1c* genes using ggtree v3.8.2 [42] and displayed other results using the ggplot2 v.3.5.0 R package [43]. The MCC trees for all gene segments were also visualized through Nextstrain’s Auspice platform [44] and are available at https://nextstrain.org/community/NakarinP/TRV, accessed on 28 July 2025.

## 3. Results

### 3.1. Ancestral Dates, Evolutionary Rates, and Population Dynamics

Bayesian phylodynamic analysis of ten TRV gene segments estimated the common ancestor of TRVs in the U.S. to be between the late 1980s and the early 2000s. However, dates of emergence estimated from tMRCAs (time to the most recent common ancestors) varied by segment. For the *L* and *M* class segments, the median tMRCA ranged from 1986 to 1992 (95% highest posterior density [HPD] range: 1975–1998). On the other hand, tMRCAs of most *S* class segments (*S2*, *S3*, and *S4*) were inferred to be later, ranging from 1992 to 2000 (95% HPD range: 1983–2005). The tMRCA for the σC protein coding region of the *S1* segment (*S1c*) was substantially more recent than the rest of the TRV genome, approximated to have emerged in 2006 with a narrow confidence interval (95% HPD: 2004–2007), which is just a year before the earliest sample date (Figure 1A).

Nucleotide substitution rates of TRV were relatively consistent among different gene segments. The median rates and corresponding 95% HPDs of seven out of ten segments (all *L* class segments, *M1*, *M3*, *S2*, and *S4*) were between 1 × 10^−3^ to 2 × 10^−3^ substitutions per nucleotide site per year (s/n/y). The *S3* segment and *S1c* gene were estimated to evolve slightly faster, with median rates between 2 × 10^−3^ and 3 × 10^−3^ s/n/y. Interestingly, the *M2* segment evolved more than twice as fast as the other nine segments, with a median rate of 4.6 × 10^−3^ s/n/y (95% HPD = 3.8 × 10^−3^–5.4 × 10^−3^ s/n/y) (Figure 1B).

Although the ancestral date and evolutionary rate vary by gene segment, Bayesian Skygrid analysis revealed similar trends in the temporal dynamics of the TRV effective population size in the U.S. The median effective population size of most segments gradually increased from their origin in the 1980s, reaching the first and highest peak in the early 2010s, with a population size about 10 times higher than at the beginning. After 2015, the virus population sharply decreased, rebounded to a smaller peak in 2019, and then continuously declined until the most recent sampling point. The 95% HPD of the effective population size before the first peak is very wide, indicating uncertainty in population size estimation during the period without sequencing data. Unique trends were observed for some segments, including a recession in the late 2000s and recovery in 2010 of the *M2* population before the first peak, as well as a rapid population expansion following the emergence of *S1c*, which contributed to the first peak (Figure 1C).

### 3.2. Spread Patterns Among Turkey-Producing Regions

Inter-regional spread of TRV was inferred using median absolute transition rates between regions, summarized across the posterior tree distribution. These rates, expressed as transitions per lineage per year, reflect the expected frequency of ancestral state changes along the phylogeny and serve as a proxy for the magnitude of viral movement from one region to another. While the example MCC trees (Figure 2A,B) partially illustrate these dispersal patterns, the rates were not directly measured from the trees but derived from the CTMC model implemented in BEAST. Only transitions with Bayes factors (BF) greater than three were considered, indicating at least moderate statistical support for the transition rate.

The most apparent dispersal pathway was from the Eastern United States to Minnesota, with estimated transition rates ranging from 0.09 to 0.16 transitions/lineage/year across different gene segments (BF > 100), except for segments *M3* and *S4*, which showed lower rates with weaker support. In contrast, the *S1c* gene segment revealed additional distinct patterns, including spread from Canada to the Eastern United States with an estimated transition rate of 0.23 transitions/lineage/year (BF 30–100) and from Minnesota to the Dakotas with a transition rate of 0.14 transitions/lineage/year (BF > 100) (Figure 2C and Appendix A).

Overall, Minnesota, the Eastern United States, and Canada emerged as common sources of viral spread to other regions. Minnesota acted as a source for a median of five other regions (with a median outgoing transition rate of 0.06 transitions/lineage/year) across the ten gene segments. The Eastern U.S. was the source for a median of four regions (0.06 transitions/lineage/year), and Canada for a median of 3.5 regions (0.05 transitions/lineage/year). Most regions acquired the virus from a median of two to three other regions, with incoming transition rates ranging from 0.03 to 0.06 transitions/lineage/year, except for Minnesota, which had a higher incoming rate of 0.09 transitions/lineage/year (Appendix A).

### 3.3. Spread Patterns Between Clients

Focusing on the client groups within our dataset, the discrete trait analysis of all ten gene segments consistently indicated that client “a”—breeder turkey producer contributing 25.6% of total samples—served as a major source of viral spread to the “u” group (37% of total samples). The “u” group comprised multiple small clients (commercial turkey producers), each contributing less than 5% of the samples. Median transition rates from client “a” to the “u” group ranged from 0.11 to 0.18 transitions/lineage/year (BF >100). Conversely, the “u” group also transmitted the virus back to client “a” to a lesser extent, with median transition rates ranging from 0.04 to 0.14 transitions/lineage/year (BF 10 to >100) across all segments. Client “c,” a commercial turkey producer, contributed 10% of the total samples. This client was identified as an important source of between-client spread. Specifically, the *S2*, *S3*, and *S4* segments estimated median transition rates of 0.12–0.17 transitions/lineage/year (BF 10–100) from client “c” to client “a,” while the *S3* and *M2* segments inferred rates of 0.11 and 0.18 transitions/lineage/year, respectively, from client “c” to the “u” group. The MCC trees for representative segments (Figure 3A,B) illustrate parts of these inferred pathways, while the overall client-level spread network, summarized across the posterior distribution of all segments, is shown in Figure 3C and Appendix A.

Across the analyzed clients, there was substantial variation in the inferred spread network. The “u” group exhibited the highest median number of recipient clients (four) and source clients (three), as well as the highest median incoming transition rate (0.10 transitions/lineage/year). In contrast, clients “d” and “e” (each contributing 5.7% of total samples) showed the lowest median number of recipients (one) and generally lower median transition rates (~0.03 transitions/lineage/year). Clients “a,” “b,” and “c” fell in the mid-range for most metrics. Notably, client “a” stood out with a relatively high median number of recipient clients (3.5) and a median outgoing transition rate of 0.07 transitions/lineage/year, while client “b” (16.1% of total samples) had the lowest median number of source clients (one) (Appendix A).

### 3.4. Patterns of Pathotype Shift

The frequency of TRV pathotype shifts was inferred using the median ancestral state transition rate between pathotypes estimated by discrete trait analysis. With over 70% of the total samples, TARV was identified as the ancestral pathotype for both unclassified TRV (0.04–0.09 transitions/lineage/year in most segments except *L1* and *L3*, BF > 100) and THRV (0.05–0.06 transitions/lineage/year, BF 3–100 across four segments: *M3*, *S2*, *S3*, and *S4*). In contrast, TARV occasionally emerged from clades with mostly THRV and unclassified TRV, with estimated median transition rates of 0.04–0.11 transitions/lineage/year in the *S1c* and *M2* segments (BF values between 3 and >100). For TERV, which comprised less than 2% of total samples, only infrequent transitions with other pathotypes were detected. The most notable link was observed in the *S1c* and *S4* segments, where the rate was 0.04 transitions/lineage/year from TERV to TARV (BF 3–30). The MCC trees for representative segments (Figure 4A,B) illustrate examples of these inferred pathotype shifts, while the overall pattern across all segments is summarized as a network in Figure 4C and Appendix A. Across all segments, the median transition rates—either incoming or outgoing—between any pair of pathotypes did not exceed 0.06 transitions/lineage/year, indicating a relatively low frequency of pathotype shifts compared to inter-regional and inter-client spread (Appendix A). The lack of clustering of pathotypes within the tree also suggests that specific pathotypes emerged repeatedly within the virus’s evolutionary history, as opposed to certain pathotypes being associated with specific viral clades.

## 4. Discussion

In this study, we utilized nationally representative genome sequencing data of turkey reovirus (TRV) to estimate its historical spread and evolutionary patterns across various dimensions, including geographic regions, client groups, and pathotypes. Our analyses across ten genomic segments revealed both segment-level consistencies and variability that provided insight into the potential genomic reassortment events, which resulted in the emergence and re-emergence of the virus. Notable consistent patterns across most segments included frequent viral spread from Eastern states to Minnesota, transmission from a major breeder client (who comprised the largest number of samples) to a group of smallholders, and a pathotype shift from arthritis-causing strains (TARV) to those associated with hepatitis (THRV) and unclassified pathotypes (TRV). Interestingly, the *S1c* gene yielded distinct results in several analyses, such as exhibiting a higher evolutionary rate, a more recent ancestor, a unique transmission pattern from Canada to the Eastern U.S. states, and a reverse pathotype shift from THRV and unclassified TRV back to TARV.

The estimated population dynamics from most gene segments showed a gradual increase from the earliest time points, peaking in the mid-2010s, followed by a steady decline over the past decade—a pattern also observed in other studies of ARV dynamics [45]. These results align with the known epidemiology of TRV in the U.S., where the virus was first isolated from turkeys in 1980 [46] and then re-emerged alongside other ARV genotypes during the 2010s [9]. In addition to the introduction of autogenous vaccines in breeder turkeys in the early 2010s in response to outbreaks [47] and improvements in farming systems, a reduction in sequencing efforts—largely driven by academic and research-focused initiatives—may also explain the observed decline in the estimated population size of both TRV and ARV, reflecting fluctuating research interest in the disease over time.

The origin of TRV in the U.S. was estimated to be relatively recent—around the 1980s—compared to the global ARV population and the specific genotype to which U.S. TRV belongs, the most recent common ancestors of which were dated back to the 14th and 18th centuries, respectively, based on the *S1c* gene [45]. Indeed, the tMRCA for U.S. TRVs based on the *S1c* gene was estimated to be as recent as 2006. This finding suggests that all TRVs in this study descended from a reassortment event involving a novel *S1c* segment that was absent in the turkey population during the 1980s and 1990s. The acquisition of this segment may have conferred increased epidemic potential, either through a selective sweep or spillover from another avian host. Intriguingly, this inferred reassortment event was temporally aligned with the re-emergence of TRV as a pathogenic issue in turkeys in 2011. Similar interspecies reassortment has also been reported in duck reoviruses, where the *S3* and *S4* gene segments were derived from chicken ARVs, suggesting that the mechanisms and patterns of interspecies reassortment can vary among avian reoviruses [48].

Supporting this evolutionary transition, phylogenetic analyses have shown close relationships between the *S1c* genes of turkey-origin ARVs (TARVs) and chicken-origin strains such as GEL13a98M, suggesting possible reassortment between viruses associated with malabsorption syndrome and viral arthritis [24]. A 2008 TERV isolate (TERV2) was also shown to replicate in turkey poults’ tendons by 21 days post-infection and induce tenosynovitis lesions. Although milder than those caused by TARVs, the lesions indicate that TERV2 may represent an evolutionary link between earlier, less virulent TERVs and the more pathogenic TARVs that emerged after 2010 [4].

A sharp increase in TRV population size, detected only from the *S1c* gene during the 2010s, along with its elevated evolutionary rate relative to most other genes, suggests that a reassortment involving a novel *S1c* segment likely played a key role in the re-emergence of both ARV and TRV [9,13,15]. Notably, the *M2* gene, which encodes the outer capsid protein µB critical for host cell entry [6], exhibited the highest evolutionary rate among all segments and showed a unique decline in population size during the re-emergence period. This pattern is consistent with multiple studies indicating that *M2* is either the most or second most variable gene after *S1* [13,18,24,49], and is frequently involved in reassortment events, including evidence of gene exchange between turkey and chicken ARVs [12,13]. The reassortment of *S1c* and *M2* was also clearly reflected in our discrete trait analyses, where the inferred transmission patterns for these two segments were consistently distinct from those of other genomic regions across multiple dimensions, i.e., geographic location, client, and pathotype.

Phylogeographic analysis of eight out of ten viral genes identified a common pathway of TRV spread from the Eastern states to Minnesota. As of 2024, these regions have the highest turkey production in the U.S., with 50.7 and 33.5 million heads in Eastern states and Minnesota, respectively [50]. Such frequent long-distance TRV spread is unsurprising given the U.S. turkey production system structure, where large companies operate multiple integrated complexes with multi-stage production (breeders, hatcheries, brooding) but typically use contract producers for finishing, with premises from all stages potentially located across different areas or states [51]. Moreover, corn and soybeans—main ingredients comprising over 80% of commercial turkey feed [51]—are predominantly produced in the Midwest (corn belt) [52]. Given these factors, interregional live animal and feed movements, plus other transportation between premises, are unavoidable and can lead to long-distance spread through direct contact with infected animals or through fomites such as contaminated feed, wood shavings, or feathers [53,54], where ARV particles survive extended periods even under extreme conditions like disinfectants or heat [55]. This rationale also applies to the strong spread pattern from Canada to the U.S. Eastern states captured only by the *S1c* gene, since Canada hosts a major breeding company that distributes poults for commercial turkey production to multiple U.S. turkey-raising areas. Unfortunately, there is very limited to no publicly available data on patterns of turkey movement in the U.S.

The narrative of TRV spread, following the vertically integrated company structure from top (breeders) to bottom (finishers), became clearer when phylogeography was discretized by client groups. Client “a,” a turkey breeding producer who contributed the largest set of TRV samples, emerged as a strong source of transmission to smallholder groups such as “u.” In contrast, client “b,” another breeder company and the second-largest contributor of samples after “a,” showed no apparent TRV spread connections to other groups. This could suggest that client “b” either had more effective TRV control measures in place, preventing downstream transmission, or operated within a production system that lacked epidemiological connections to other herds represented in this study. Potential genome reassortments were indicated by gene-specific transitions from client “c” to both client “a” and group “u.” This pattern suggests that client “c” maintained connections with both breeders and smallholders—consistent with its identity as a commercial integrated company that likely purchases poults from a primary breeder and finishes turkeys at other contract producers [51,56]. Notably, evidence of upstream (counter-flow) transmission was also detected between groups otherwise linked by downstream spread, highlighting the critical role of indirect transmission [53,54] in driving national-scale outbreaks.

Characterizing the shift in TRV pathotypes poses unique challenges, given that diagnostic submissions are inherently biased toward samples from turkeys with overt clinical signs. As a result, TARV, which causes the most characteristic arthritis lesions, was overrepresented in this study. In contrast, TERV—believed to be the progenitor of other pathotypes, given that the intestine is the primary site of TRV entry and replication [57], and that enteritis was the predominant clinical sign before the 2011 lameness outbreak [4]—accounted for only 2% of total samples and was mostly detected before 2010 (Appendix A). This likely contributed to its limited genetic relatedness to samples collected after the outbreak. Furthermore, TERV pathogenesis is inconsistent, ranging from subclinical infection to mild or severe enteritis [58], making it harder to detect.

Despite these limitations, our analysis still captured the major pathotype transitions—from TARV to THRV and unclassified TRV. THRV was first identified in 2019 as a causative agent of hepatitis in turkeys and was hypothesized to have evolved from TARV [4]. Our findings support this hypothesis, showing that at least four of the ten genomic segments in THRV are frequently derived from TARV. In the case of unclassified TRV strains isolated from the heart or spleen, we observed that eight segments often originated from TARV. Previous studies also reported that a specific TARV subgroup—TARV-O’Neil—consistently caused histologic heart lesions, even though gross epicarditis or myocarditis was variable and sometimes absent [4]. The observed reversion of THRV and unclassified TRV back to TARV-like genotypes based on *S1c* and *M* gene analyses is not unexpected. Different lesion types can result from a single infection, depending on the timing of peak viral replication in each tissue. For instance, THRV can migrate from the liver to the tendon over time and highly pathogenic THRVs were reported to cause both hepatitis and arthritis in experimentally infected turkeys [4]. Similarly, THRV from a two-week-old turkey poult caused hepatitis, myocarditis, and tenosynovitis in day-old chickens on experimental infection [59].

As mentioned above, the main limitations of our phylodynamic analyses stem from sampling bias associated with submissions to the Minnesota Veterinary Diagnostic Laboratory (MVDL). These submissions may partially reflect disease incidence but were also geographically biased toward certain regions, as the MVDL is located in the Midwestern United States. Ratios of % TRV genome sample size to % 2024 turkey population by region (Appendix A [60,61]) indicate that some regions were overrepresented (e.g., Dakotas) and others underrepresented (e.g., Midwest) relative to turkey population size. Because turkey population density may not directly correlate with outbreak incidence, which is undocumented and varied through time, and the overall dataset is small, subsampling was not appropriate as it would further reduce analytical power. We also note the absence of sequences from the Western states in our dataset, as California alone raises ~6.2 million turkeys; this limits our ability to assess potential connections between the Western states and other regions. Additionally, key metadata—such as production type, herd size, geographic location, and client identification—are confidential, which restricted our ability to conduct more comprehensive analyses or interpret the findings beyond what has been described. Nevertheless, our study offers several important insights that align with known characteristics of the U.S. turkey production industry and previous research. These insights can help industry stakeholders better understand the national and historical landscape of TRV, which may be valuable for informing large-scale disease control strategies.

Interpreting phylodynamic results is particularly challenging for segmented genome viruses, the evolutionary dynamics of which are heavily influenced by genomic reassortment—i.e., the horizontal exchange of genome segments between viruses co-infecting the same host. For example, the global dispersal pattern of avian reoviruses (ARVs) has been inferred exclusively from *S1c* gene sequences (referred to as σC in the original study) [45]. In comparison, our work on TRV revealed that spread patterns inferred solely from *S1c* differed substantially from those based on other genomic segments. This suggests that if full-genome data had been available, the conclusions of such an ARV study might have differed. Due to unequal sequence availability and sequencing efforts across genome segments of segmented viruses, many phylogeographic studies rely on a single, abundantly sequenced gene—such as the *HA* gene for influenza viruses or the *S1c* gene for ARVs—focusing on subtype-specific spread or evolutionary patterns based on that gene [18,62,63]. Our study is among the few [64,65] that have inferred the spatial distribution of a segmented virus using multiple genomic segments, providing broader insights by accounting for the effects of reassortment.

## 5. Conclusions

The phylodynamic analyses conducted in this study integrated the fragmented history of turkey reovirus (TRV) epidemiology in the United States into a coherent narrative—tracing its trajectory from the first identification in turkeys during the 1980s to the post-TARV emergence after 2011 and into the present. This overarching history was reconstructed using comprehensive whole-genome analysis, revealing temporal patterns in population dynamics and evolutionary rates. Notably, the *S1c* and *M2* segments exhibited slightly different dynamics and faster evolutionary rates, suggesting potential reassortment with other co-emerging avian reoviruses. The inferred national spread patterns of TRV revealed a dominant transmission route linking the two largest turkey-producing regions—from the Eastern states to Minnesota—likely driven by inter-regional movement of animals or feed. Additional major routes, such as from Canada to the Eastern U.S. and from breeders to smallholders, illustrated both downstream transmission along the multi-stage turkey production system and possible upstream spread through indirect routes. Finally, we observed that evolutionary pathotype shifts occurred more frequently from the typical arthritis-associated form to those causing hepatitis or cardiac infections. These insights can inform national TRV control strategies and support long-term disease monitoring efforts by stakeholders in the turkey industry.

## Figures and Tables

**Figure 1 viruses-17-01185-f001:**
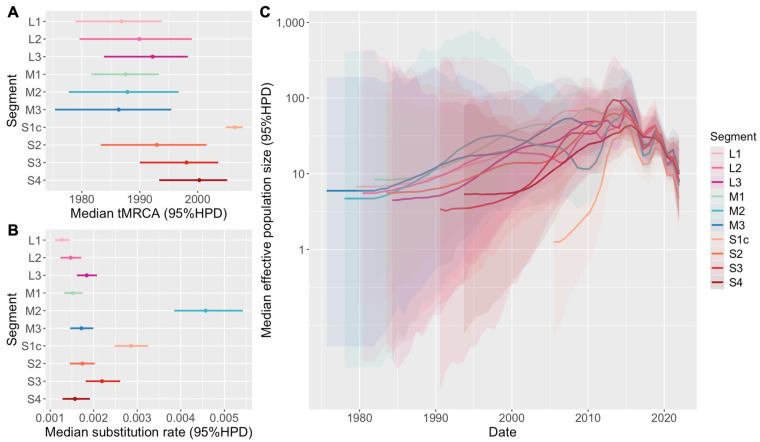
Phylodynamic estimates based on ten TRV genome segments. (**A**). Estimated median time to the most recent common ancestor (tMRCA) for each segment, with corresponding 95% high posterior density (HPD) intervals. (**B**). Estimated median nucleotide substitution rates (substitutions per site per year) with 95% HPD intervals across the ten genome segments. (**C**). Temporal dynamics of the median effective population size (Ne) inferred from each segment, with shaded regions indicating 95% HPD intervals.

**Figure 2 viruses-17-01185-f002:**
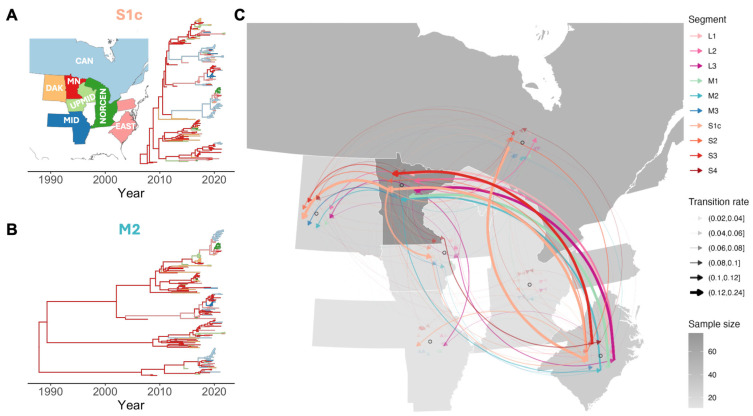
Inferred historical inter-regional spread of TRV in the U.S. and Canada. Maximum clade credibility (MCC) trees of the *S1c* (**A**) and *M2* (**B**) segments are shown, with branches colored according to the most probable geographic region of origin. The regional color legend is provided as a reference map in the top left of panel (A), with the following abbreviations: CAN (Canada), DAK (Dakotas), MN (Minnesota), UPMID (Upper Midwest), MID (Midwest), NORCEN (North Central), and EAST (East). (**C**). Summary of inferred inter-regional transition rate from all gene segments. Arrows represent inferred median viral transition rate (transitions/lineage/year) between regions, colored by the TRV genome segment used for estimation. Arrow width is proportional to the transition rate. Regional colors on the map indicate the number of samples collected from each area in this study. Data are based on BSSVS discrete trait analysis, showing only transition rates with Bayes factor > 3.

**Figure 3 viruses-17-01185-f003:**
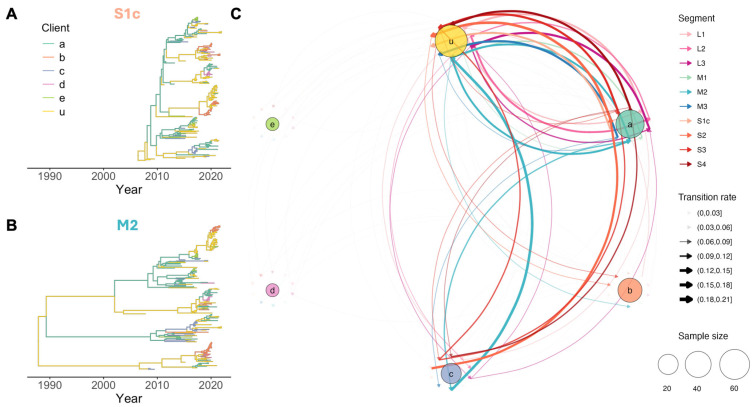
Inferred between-client TRV spread. Maximum clade credibility (MCC) trees for the *S1c* (**A**) and *M2* (**B**) segments, with branches colored by the client with the highest posterior probability (color legend shown in the top left of A; client types: a = breeder/producer, b–e = commercial producers, u = small commercial producers [pooled]). (**C**) Summary network of inferred between-client spread across all gene segments. Arrows represent median transition rate (transitions/lineage/year) between clients, colored by the TRV genome segment used for estimation. The arrow width is proportional to the transition rate, while the circle size indicates the number of samples collected from each client. Estimates are based on BSSVS discrete trait phylogeographic analysis, showing only transition rates with a Bayes factor of >3.

**Figure 4 viruses-17-01185-f004:**
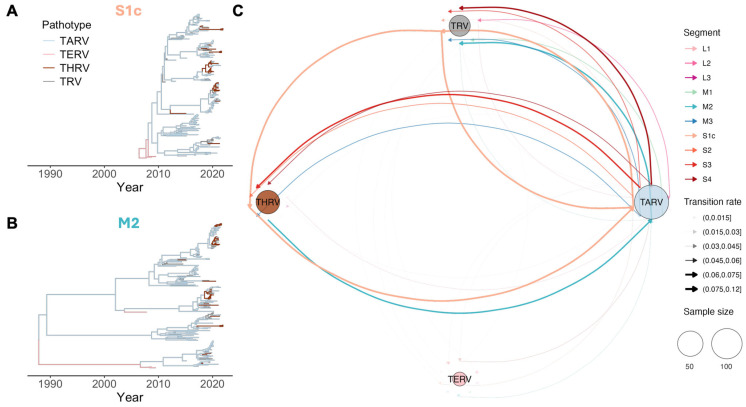
Inferred pathotype shifts of TRV. Maximum clade credibility (MCC) trees for the *S1c* (**A**) and *M2* (**B**) segments, with branches colored by the pathotype with the highest posterior probability (color legend shown in the top left of (**A**). (**C**). Summary network of inferred pathotype shifts across all gene segments. Arrows represent median transition rate (transitions/lineage/year) between pathotypes, colored according to the TRV genome segment used for estimation. Arrow width is proportional to the transition rate, and circle size indicates the number of samples associated with each pathotype. Estimates are based on BSSVS discrete trait phylogeographic analysis, displaying only transition rates supported by a Bayes factor > 3.

## Data Availability

The MCC phylogenetic trees for all gene segments are publicly available at https://nextstrain.org/community/NakarinP/TRV, accessed on 27 July 2025. Sequencing data owned by MVDL remains confidential. Additional data supporting the findings of this study are available from the corresponding author upon request.

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
