# Peer review of "Connecting the Evolution and Spread of Turkey Reovirus Across the United States: A Genomic Perspective"

_viruses, 2025, doi:10.3390/v17091185_

Round 1
Reviewer 1 Report
Comments and Suggestions for Authors
This study is the first to reveal the core role of S1c and M2 genes in TRV reassortment and re-emergence through segmented genome analysis. It has uncovered a complex transmission network via multi-dimensional phylogeographic analyses, identified TRV transmission hotspots and key control nodes, and provided targets for vaccine development and biosecurity strategies. However, there are still some details and issues that need improvement in this article. The major concerns are listed as follows:
- A total of 87 out of 211 samples (41.2%) in this study were collected from Minnesota (the location of MVDL), while samples from other regions are scarce (e.g., only 20 samples from the Midwest). Although multiple regions are covered, there may be geographic sampling bias. It is suggested to supplement explanations on how to correct for geographic bias through statistical methods, or discuss the potential impact of unincluded regions on the overall conclusions.
- The study indicates that the S1c and M2 segments have higher evolutionary rates, suggesting possible reassortment with other avian reoviruses, which coincides with the re-emergence of TRV in 2011. However, the specific molecular mechanisms of the reassortment events are not clarified. It is recommended to supplement detailed sequence analysis of the reassortment regions and homology comparisons with known strains.
- The map in Figure 2 is not annotated with state boundaries , and it is suggested to add state abbreviations. The labels (a, b, c, d, e, u) in Figure 3C lack clear identity explanations; it is recommended to supplement labels for client types in the legend or directly annotate the meanings of the abbreviations next to the nodes.
- Previously study reported that the S3/S4 gene segments of duck reoviruses (DRV)were derived from chicken ARVs, indicating that DRVs can acquire new gene segments through interspecies reassortment (PMID: 39915856 ). The finding is similar to the evolutionary patterns of turkey reovirus (TRV) in this study. It is suggested to discuss the similarity of ARV, TRV and DRV emphatically when discussing reassortment mechanisms.
Author Response
Comment 1: A total of 87 out of 211 samples (41.2%) in this study were collected from Minnesota (the location of MVDL), while samples from other regions are scarce (e.g., only 20 samples from the Midwest). Although multiple regions are covered, there may be geographic sampling bias. It is suggested to supplement explanations on how to correct for geographic bias through statistical methods, or discuss the potential impact of unincluded regions on the overall conclusions.
Response 1: We thank the reviewer for this important comment. Our ratios represent % WGS sample size relative to % regional turkey population (Supplementary Table S6), showing overrepresentation in some regions (e.g., Dakotas) and underrepresentation in others (e.g., Midwest). While this highlights sampling disparity, turkey population density does not necessarily reflect outbreak incidence, and subsampling in relatively small datasets is unlikely to improve inference. Importantly, our dataset does not include sequences from California, which raises ~6.2 million turkeys, limiting our ability to evaluate potential connections between the western U.S. and other regions. These points have been added to the discussion (Lines 487–499).
Comment 2: The study indicates that the S1c and M2 segments have higher evolutionary rates, suggesting possible reassortment with other avian reoviruses, which coincides with the re-emergence of TRV in 2011. However, the specific molecular mechanisms of the reassortment events are not clarified. It is recommended to supplement detailed sequence analysis of the reassortment regions and homology comparisons with known strains.
Response 2: We appreciate the reviewer’s insightful suggestion. The primary objective of our study was to investigate the population dynamics and spatial spread of TRV, rather than to directly assess reassortment mechanisms. Therefore, detailed reassortment analyses were not included in the results section. Nevertheless, the higher evolutionary rates observed in the S1c and M2 segments were interpreted in the context of previous studies, which suggested reassortment events. Our inferred evolutionary rates and effective population size trajectories are consistent with those prior findings, reinforcing the validity of our conclusions. Since detailed phylogenetic analysis of reassortment has already been addressed by prior work (https://doi.org/10.1016/j.meegid.2015.03.029), and such analyses fall outside the scope of our study, we believe additional reassortment-focused analyses would not substantially strengthen the aims of this manuscript.
Comment 3: The map in Figure 2 is not annotated with state boundaries, and it is suggested to add state abbreviations. The labels (a, b, c, d, e, u) in Figure 3C lack clear identity explanations; it is recommended to supplement labels for client types in the legend or directly annotate the meanings of the abbreviations next to the nodes.
Response 3: We thank the reviewer for this helpful suggestion. We have added region abbreviations to the colored map legend in Figure 2, along with full region explanations in the figure description. Additionally, the client types corresponding to each label (a, b, c, d, e, u) in Figure 3C are now clearly explained in the figure description.
Comment 4: A previous study reported that the S3/S4 gene segments of duck reoviruses (DRV)were derived from chicken ARVs, indicating that DRVs can acquire new gene segments through interspecies reassortment (PMID: 39915856 ). The finding is similar to the evolutionary patterns of turkey reovirus (TRV) in this study. It is suggested to discuss the similarity of ARV, TRV, and DRV emphatically when discussing reassortment mechanisms.
Response 4: We thank the reviewer for this valuable suggestion. This is indeed a strong comparative example of interspecies gene reassortment in reoviruses. We apologize for missing it in the original version. We have now included this discussion in the revised manuscript (Lines 391–394), as recommended.
Reviewer 2 Report
Comments and Suggestions for Authors
The report by Pamornchainavakul et al. focuses on connecting the evolution and spread of turkey reovirus across the United States: A Genomic Perspective. This work provides extensive evaluation of WRV. However, the methods used and findings are not novel. Listed below are some observations that the authors should consider in the presentation, interpretation, and correction of some wrong description or citation.
- Lines 51-52: For example, σC, one of the products of the S1 gene widely employed for ARV classification, is the most variable protein, which mediates virus, cell attachment and is a target of neutralizing antibodies. The sigma C of ARV was found to be an apoptosis inducer (Virology 321: 65-74, 2004). Although the author describes some of the biological function of sigma C of ARV, but did not describe that it is an apoptosis inducer. Additional explanation in the Introduction should be given.
- Line 53-55: Six ARV genotypes have been described based on the S1 gene or σC amino acid phylogenies, and most reoviruses found in turkeys have been classified as genotype (29,13,22). The earliest research by Liu et al ( Virology 314: 336-349, 2003) confirmed that sigma C protein has the largest variation, which is used for genotyping. The authors did not cite the earliest literature and its contributions.
- Wrong citation or missing citation: Six ARV genotypes have been described based on the S1 gene or σC amino acid phylogenies, and most reoviruses 54 found in turkeys have been classified as genotype (29,13,22). The earliest/ first report by Liu et al ( Virology 314: 336-349, 2003) confirmed that sigma C protein has the largest variation, which is used for genotyping. The authors did not cite the earliest literature and its contributions.
- How many genotypes of turkey reovirus were found? The authors need to further discuss this issue, but not just showing the phylogenetic tree.
5 An earlier report suggesting that genetic diversity and reassortment of the S-class genome of ARV segments and multiple cocirculating lineages and synonymous substitutions predominate over nonsynonymous in the S-class genes, even though genetic diversity and substitution rates vary among the viruses. The authors need to analyze whether the sigma C-encoding gene occurs nonsynonymous substitutions predominate over synonymous.
6.The quality of Fig. 2A-2B, 3A-3B, and 4A-4B is poor. The phylogentic trees are not clear and hard to read.
- How to PCR or cloning of all TRV genomes for sequencing and phylogentic analysis ?
- Reference 17: the title of this report is missing.
Author Response
Overall Response: We thank the reviewer for the kind summary. The methods we used in this study, though it is not novel but is a standard advance approach researchers employ to estimate historical pathogen spread. Moreover, we believe the findings of our work provide novel contributions, as this is the first assessment of the spread patterns of turkey reovirus (TRV, not WRV or waterfowl reovirus) within the U.S. using complete, nationally representative genome data (beyond sigma C alone). We appreciate the reviewer’s suggestion to improve the English in the manuscript. However, we would like to assure that the manuscript already meets language standards, as it has been revised multiple times by a native American English speaker. Please also see our detailed responses to each comment below.
Comment 1: Lines 51-52: For example, σC, one of the products of the S1 gene widely employed for ARV classification, is the most variable protein, which mediates virus-cell attachment and is a target of neutralizing antibodies. The sigma C of ARV was found to be an apoptosis inducer (Virology 321: 65-74, 2004). Although the author describes some of the biological functions of sigma C of ARV, but did not describe that it is an apoptosis inducer. An additional explanation in the Introduction should be given.
Response 1: We thank the reviewer for pointing out this important function of the sigma C protein. We have now added the information regarding its role as an apoptosis inducer, along with the appropriate citation, to the Introduction (Line 63).
Comment 2: Line 53-55: Six ARV genotypes have been described based on the S1 gene or σC amino acid phylogenies, and most reoviruses found in turkeys have been classified as genotype (29,13,22). The earliest research by Liu et al ( Virology 314: 336-349, 2003) confirmed that the sigma C protein has the largest variation, which is used for genotyping. The authors did not cite the earliest literature and its contributions.
Response 2: We thank the reviewer for raising this point. In this sentence, we cited the three papers (9, 13, 22) because they specifically show the phylogenetic grouping of turkey-origin reoviruses as “genotype 2.” While Liu et al. (2003) did not display the phylogenetic placement of turkey-origin reoviruses, the study did initiate the use of sigma C protein for phylogenetic reconstruction of ARVs. Therefore, we have now cited Liu et al. after the statement “Six ARV genotypes have been described based on the S1 gene or σC amino acid phylogenies” (Line 65) to acknowledge their contribution.
Comment 3: Wrong citation or missing citation: Six ARV genotypes have been described based on the S1 gene or σC amino acid phylogenies, and most reoviruses 54 found in turkeys have been classified as genotype (29,13,22). The earliest/ first report by Liu et al ( Virology 314: 336-349, 2003) confirmed that the sigma C protein has the largest variation, which is used for genotyping. The authors did not cite the earliest literature and its contributions.
Response 3: This comment overlaps with Comment 2. Please see our revised response above.
Comment 4: How many genotypes of turkey reovirus were found? The authors need to further discuss this issue, but not just by showing the phylogenetic tree.
Response 4: All turkey reovirus samples in this study were identified as avian reovirus genotype 2 based on the sigma C phylogenetic tree, and this information has been added to Lines 66–67 and Supplementary Figure S5. However, if the reviewer was referring to sub-genotypes within the turkey reovirus samples, we did not emphasize such detailed classifications for the following reasons: (1) several previous studies have already described turkey reovirus genotypes using different gene segments (as noted in Line 60, e.g., Mor et al., 2014–2015); and (2) our estimation of TRV historical spread was based on ancestral state transitions along the Bayesian MCC phylogenetic tree, independent of genotype classifications, which can vary depending on the gene segment analyzed and the distance thresholds applied in each study. The phylogenetic trees presented in Figures 2–4 are intended to illustrate ancestral state transitions across the MCC trees, rather than to define or compare genotypes.
Comment 5: An earlier report suggested that genetic diversity and reassortment of the S-class genome of ARV segments and multiple cocirculating lineages and synonymous substitutions predominate over nonsynonymous in the S-class genes, even though genetic diversity and substitution rates vary among the viruses. The authors need to analyze whether the sigma C-encoding gene occurs with nonsynonymous substitutions predominating over synonymous.
Response 5: We appreciate this thoughtful suggestion. We conducted a selection pressure analysis using the mixed effects model of evolution (MEME) (https://doi.org/10.1371/journal.pgen.1002764). The results indicated that only four codons (positions 118, 124, 147, and 213) out of 140 in the σC protein were under significant diversifying selection (non-synonymous > synonymous substitutions). However, we chose not to include this analysis in the manuscript, as it does not directly contribute to the central objective of our study, which is to estimate the historical spread of TRV.
Comment 6: The quality of Fig. 2A-2B, 3A-3B, and 4A-4B is poor. The phylogenetic trees are not clear and hard to read.
Response 6: We apologize that the phylogenetic trees appear unclear. These figures were intended to provide a brief overview of ancestral state transitions across MCC trees, which support the historical spread patterns shown in Figures 2C, 3C, and 4C. The resolution may also appear reduced due to the way figures are automatically rendered within the journal platform; for higher-quality versions, please refer to the figures we submitted separately. For readers interested in exploring the MCC trees of all gene segments in detail, we have also provided a hyperlink to our Nextstrain visualization, which enables interactive exploration at high resolution.
Comment 7: How to PCR or cloning of all TRV genomes for sequencing and phylogenetic analysis?
Response 7: The samples analyzed in this study were collected over more than a decade, during which RNA amplification and sequencing technologies varied. To address this, we included a general overview of these methods in Lines 102–105: “RNA extraction, amplification, and sequencing varied over time but generally involved RT-PCR followed by sequencing, using Sanger technology for earlier samples (University of Minnesota Genomic Center) and Illumina technology for more recent samples (Illumina, San Diego, CA, USA).”
Comment 8: Reference 17: The title of this report is missing.
Response 8: We thank the reviewer for catching this error. The missing title in Reference 17 has now been corrected.
Round 2
Reviewer 2 Report
Comments and Suggestions for Authors
Accepted for publication